# The Evolving Interplay of SBRT and the Immune System, along with Future Directions in the Field

**DOI:** 10.3390/cancers14184530

**Published:** 2022-09-19

**Authors:** Mihailo Miljanic, Steven Montalvo, Maureen Aliru, Tidie Song, Maria Leon-Camarena, Kevin Innella, Dragan Vujovic, Ritsuko Komaki, Puneeth Iyengar

**Affiliations:** 1Department of Radiation Oncology, University of Texas Southwestern Medical Center, Dallas, TX 75390, USA; 2Department of Internal Medicine, University of Texas at Austin, Austin, TX 78705, USA; 3Philadelphia College of Osteopathic Medicine, Philadelphia, PA 19131, USA; 4Icahn School of Medicine at Mount Sinai, The Mount Sinai Hospital, New York, NY 10029, USA; 5Emeritus Professor of Radiation Oncology, UT MDACC, Adjunct Professor of Radiation Oncology Baylor College of Medicine, Houston, TX 77030, USA

**Keywords:** SBRT, SAbR, PULSAR, adaptive radiation therapy, immunotherapy in combination with radiation therapy, SBRT and immunotherapy

## Abstract

**Simple Summary:**

We provide this commentary of stereotactic body radiotherapy (SBRT), and describe our evolving understanding of this treatment approach, its effects on the immune system, and the ability to stimulate immune cells to further recognize and attack cancer. The aim of this work is to describe our current knowledge of how SBRT effects the environment within the tumor and the immune cells present, whether timing the combination of this treatment with that of immunotherapy may have an impact on the body’s own immune response, and what the latest approaches in the field are in regards to this radiation treatment modality. Among these latest and exciting developments is Personalized Ultrafractionated Stereotactic Adaptive Radiation Therapy, known as PULSAR. This latest approach is described in detail herein, and may represent a leading novel method for adapting radiation treatments to treatment-induced tumor changes over time and stimulating the body’s immune response against tumor cells.

**Abstract:**

In this commentary, we describe the potential of highly ablative doses utilizing Stereotactic Body Radiation Therapy (SBRT) in single or few fractions to enhance immune-responsiveness, how timing of this approach in combination with immune-checkpoint inhibitors may augment treatment-effect, and whether Personalized Ultrafractionated Stereotactic Adaptive Radiation Therapy (PULSAR) is an avenue for future advancement in the continued endeavor to foster a systemic effect of therapy beyond the radiation treatment field. The ablative potential of SBRT may support an increase in tumor-antigen presentation, enhancement of immune-stimulatory components, and an improvement in tumor-microenvironment immune cell infiltration. Furthermore, the latest advancement of ablative radiation delivery is PULSAR-based therapy, whereby ablative doses are delivered in pulses of treatment that may be several weeks apart, combined with adaptive treatment to tumor changes across time. The benefits of this novel approach include the ability to optimize direct tumor control by assessment of tumor size and location via dedicated imaging acquired prior to each delivered pulse, and further potentiation of immune recognition through combination with concurrent immune-checkpoint blockade.

## 1. Introduction

It has been approximately 70 years since the term “abscopal effect” was coined to describe the effect of radiation on disease located outside of the treatment field [1], and it has been almost 2 decades since one of the first studies identified that an abscopal effect may be mediated by the immune-related sequalae of ionizing radiation, possibly leading to the response of previously untreated tumor sites elsewhere in the body [2].

Yet, a complex tumor microenvironment with immunostimulatory and inhibitory factors that can be modulated by radiation therapy (RT) has led to mixed results and often disappointing abscopal trial outcomes [3,4,5]. This is, in part, due to a misunderstanding of the tumor–immune microenvironment, the differences in immune-cell regulatory populations between different cancer types, and the effect of timing and dose on the therapy and immune-stimulation. For instance, traditionally, fractionated radiation has been implicated in the formation of an immunosuppressive environment due to the mitigation of antigen-specific T-cell populations over a course of several-weeks-long treatments that can lead to the suppression of exactly the kind of immunostimulatory response we are hoping to ignite [6]. It has more recently come into focus that not only the dose, but also the timing of radiation may be a vital step in preserving critical lymphocyte populations for the enhancement of an anti-tumor immune response. In the subsequent sections, we will therefore focus on the immunomodulatory effects of radiation on the potential of highly ablative doses utilizing stereotactic body radiation therapy (SBRT) in a single or in a few fractions to enhance immune responsivity; the way in which the timing of this approach, in combination with immune-checkpoint inhibitors, may augment the treatment effect; and whether personalized ultrafractionated stereotactic adaptive radiation therapy (PULSAR) is an avenue for future advancement in the continued endeavor to foster a systemic effect of therapy beyond the radiation treatment field.

## 2. Radiation as an Immunomodulatory Treatment Modality

Several preclinical and clinical studies have reported on the immunomodulatory effects of radiation therapy, including the stimulation of tumor-antigen release, the facilitation of antigen presentation, and the maturation and homing of T cells [7,8,9]. Furthermore, radiation can also induce immunosuppressive changes in the tumor microenvironment (TME) through the induction of transforming growth factor beta (TGFβ), indolamine 2.3-dioxygenase (IDO), and PD-L1, which results in an increase in suppressive cells in the tumor microenvironment, such as regulatory T cells (Tregs) and tumor-associated macrophages (TAMs) [10]. However, the effects of radiation therapy as an immunomodulator are determined by radiation fractionation delivery, as well as by its combination with immunotherapy. This area of research has spurred interest in the growing field of the timing of immunotherapy with radiation and the radiation-delivery schema.

Further, therapeutic radiation can have a significant effect on cytokine release, which results in immunomodulation. Cytokines that show increased expression following ionizing radiation according to several studies include TNF-alfa, IL-1, IL-6, IL-10, type 1 interferon, and TGF-beta, among others [11]. Notably, while some of these cytokines are pro-inflammatory, such as IL-1 and IL-6, others have immune-regulatory effects that dampen the immune response, such as IL-10 and TGF-beta. Therefore, the balance of immunostimulatory and pro-inflammatory cytokines against inhibitory cytokines is crucial in determining immune responsivity following radiation treatment [11].

Beyond that of cytokine release, radiation treatment has a substantial effect on the release of intracellular structures that may lead to immune recognition. One such effect is that on the release of damage-associated molecular patterns () following tumor irradiation. With tumor cell death being triggered by therapeutic radiation delivery, released DAMPs play a role in the modulation of the tumor microenvironment and can possess a range of effects that can be both immunostimulatory or immunosuppressive. Among the DAMPs released in association with radiation therapy are heat-shock proteins (HSPs), S100, and adenosine. The latter two molecules have been particularly shown to interact with both the cancer cells and the immune cells present in the tumor microenvironment, with adenosine demonstrating effects that can lead to tumor growth and the promotion of resistance to radiotherapy. Conversely, DAMPs have the capability to signal and activate the immune response, and they themselves may have different effects depending on the cell type with which they are interacting within the tumor microenvironment. Radiation therapy may further result in cyclic GMP synthase (cGAS) and stimulator of interferon gamma genes (STING) release in addition to the type-1 interferons earlier discussed, all of which can lead to the innate immune activation for recognition and cytotoxic antitumor effect [12,13].

The effects of radiation on the release of cytokines and intracellular molecules, such as DAMPs, may have both immunostimulatory and immunosuppressive effects, as does the effect of radiation on the varying cellular components of the tumor microenvironment. While radiation-induced IFN-gamma expression has been shown to promote T-cell activation directly, there are numerous effects on other cells present in the environment itself that lead to further modulation. Ionizing radiation has been shown to increase tumor-antigen presentation by present dendritic cells, which may then lead to cytotoxic T-cell activation in draining lymph nodes. Meanwhile, some studies have demonstrated an increased proportion of regulatory T-cells in the tumor following radiation treatment, and an accumulation of myeloid-derived suppressor cells (MDSCs) in the microenvironment in association with ionizing radiation, both of which lead to immunosuppressive effects. This simply demonstrates the breadth of radiation-related immunomodulatory effects, many of which have both immunostimulatory and immunosuppressive effects. With this in mind, we will next discuss how highly ablative doses of radiation utilizing SBRT may shift the immunomodulatory effects of radiation in favor of the stimulation of immune response [14].

## 3. Immunostimulatory Effects of SBRT on the Immune System

The potentially unique effect that SBRT and highly ablative doses administered in a single or a few fractions may have compared to that of traditional, conventionally fractionated radiation has received increasing attention in recent years (Figure 1). These immunopotentiating results have been demonstrated across a wide range of cancer types in both pre-clinical and clinical settings [15,16,17,18].

Following the irradiation of lung tumors, preliminary clinical studies utilizing SBRT with regimens of 60 Gy in 8 fractions or 50 Gy in 4 fractions have demonstrated increases in immunoactive CD8+ cytotoxic T-cells in the peripheral blood samples of patients detected by differences in flow cytometry, while also showing a decrease in the levels of CD4+ regulatory T-cells and granulocytic myeloid-derived suppressor cells (MDSCs), which are thought to be integral to the immunosuppressive tumor microenvironment, and which play a significant role in tumor immune-cell invasion [18,19]. Impressively, these changes were evident as early as 72 h following completion of SBRT, and they persisted for up to 6 months following therapy [19].

In pre-clinical models utilizing BALB/c mice with triple-negative breast tumors, single-dose SBRT with 12 Gy delivered in 1 fraction has been shown to enrich tumor-specific T cells and CD8+ memory cytotoxic T-cells with ablative doses, as well as to enhance the therapeutic capacity and tumor control using an immune checkpoint blockade in these models [17]. In a separate pre-clinical study utilizing B16-OVA murine melanoma models, it was observed that tumor control and tumor immunity were enhanced in a dose-dependent manner with increasing radiation doses of up to 7.5 Gy/fraction, while higher doses per fraction than this actually led to an increase in T-regulatory cell (Treg) representation [15]. Such findings indicate that there may be optimal dose regimens and fractionation schemes that allow for increased tumor control and the potentiation of immune-reactive T cells, while still remaining within a window in which Treg response remains low. This desired pattern would consist of an ideal balance of propagating highly immunostimulatory factors while minimizing immunosuppressive treatment effects.

Several clinical studies have previously shown the regression of systemic disease outside of the treatment field following SBRT. In a study of 28 patients with renal cell carcinoma who were treated utilizing this technique, with dosing regimens ranging from 30 Gy in 2 fractions to 32 Gy delivered over 4 fractions, 3 patients with metastatic disease elsewhere exhibited complete regression of the non-irradiated lesions, and they were relapse-free of systemic disease for years thereafter [20]. Here, too, it was postulated that the immunomodulation of ablative therapy may play a significant role, not only through direct cytotoxic effects, but also through the recruitment of dendritic cells, which may enhance the visibility of tumor antigens for immune cell signaling [20]. This has not been limited to this series, but it has been involved in several others as well [21,22].

Separate in vivo studies have laid a foundation for exactly this mechanism of immunoactivation through increased antigen visibility. Irradiation with a single dose of 8 Gy in 1 fraction in murine MC38 adenocarcinoma lines can increase immune recognition through the upregulation of major histocompatibility complex class 1 (MHC1) antigen-presenting molecules, as well as in Fas death receptors and in intercellular adhesion molecule (ICAM)-1 signaling following therapy [16].

While many of these findings are encouraging, mechanisms through which SBRT may augment the immune response continue to be under investigation. Furthermore, although there are now several studies showing the anti-tumor immunostimulatory potential of SBRT [23,24,25], there are still immunosuppressive effects, such as lymphopenia, that warrant consideration and further understanding regarding the effects of the dose and timing of treatments [7]. In the subsequent sections, we will further explore how the timing of SBRT in combination with immune checkpoint blockade may impact the immunomodulatory potential of ablative radiation therapy.

## 4. Timing of RT with Immune Checkpoint Blockade

Several studies have proposed that immunotherapy should be administered after radiation in order to enhance the immune response by the generated tumor antigens and to take advantage of the destroyed, pre-existing barriers of the tumor. Moreover, radiation following the activation of immune cells could abrogate the effectiveness of the antitumor cellular response [26]. Hence, delivery of immunotherapy after radiation therapy could allow for the generation of antigen-presenting cells and the effector T cells which would be readily available to respond to the tumor antigens produced during radiation. This advantage, however, is shadowed by the potential of radiation therapy to destroy the newly infiltrated T cells. As such, the optimal sequencing of SBRT would be one in which the radiation is delivered at doses that allow for vascular disruption which enhances the penetration of drugs that will augment the immunostimulatory effects, as well as enhancing the effects of the immunotherapy agent [27].

With respect to optimizing tumor lysis and immunogenicity, radiation dose delivery should maximize the systemic anti-tumor effects, while also minimizing the downstream immunosuppressive effects [27]. Preclinical and clinical studies indicate that SBRT dosing of radiation therapy in combination with immunotherapy may elicit a more robust activation of the immune system in the tumor compared to conventional fractionation dosing [28]. Kim, et al. postulated that the advantage of SBRT or SRS dosing lies in the damage done to the vasculature, as well as in the substantial release of tumor antigens from the sudden degradation of tumor cells [29]. Radiation induces rapid, direct cell death, initially, in oxygenated cells. However, with the damage to the vasculature that occurs in SBRT/SRS dosing, there is oxygenation of previously hypoxic cells, thus making them more radiosensitive. Additionally, the newly formed vasculature is abnormal and prone to leakage, allowing for the greater penetration of drugs or immune cells. This also primes the question of the timing of immunotherapy relative to SBRT.

While the timing of immunotherapy with SBRT is still an on-going area of investigation, and consensus has not been reached, several studies have reported on temporal sequences for the delivery of immunotherapy for optimal synergy. For instance, delivery of anti-CTLA4 therapy prior to radiation, or concurrently with radiation and chemotherapy, results in the increased overall survival of patients with advanced melanoma, and an abscopal effect was observed with the anti-OX40 antibody. Treatment delivery was performed 1 day following a single dose of 10 Gy of RT, and anti PD-L1 was administered on days 1–5, with radiation delivered in 2 fractions of 5 Gy [30,31,32]. Clinical trials, such as the PEMBRO-RT of patients with NSCLC study, established an improved median for progression-free survival and overall survival in the patient population that received SBRT followed by Pembrolizumab within 7 days of the last radiation dose to a single metastatic site, as compared with the group that received immunotherapy alone [33]. Moreover, KEYNOTE-001 highlights the benefit of combining radiation therapy with immunotherapy (anti-PD-1), even with RT delivery months prior to anti-PD-1 delivery [34]. Patients with advanced NSCLC who received immunotherapy following radiation had longer overall survival and progression-free survival compared to those who received only immunotherapy without radiation prior to the initiation of systemic therapy. It should be noted that the trial did not include a comparison of the converse sequencing option with radiation delivered after the administration of immunotherapy. While sequencing that involves radiation treatment followed by immunotherapy possesses a biological rationale and has gained support via the pre-clinical and clinical data discussed, concurrent administration and the amount of time between administrations still remain areas that warrant further research.

## 5. PULSAR

RT is often referred to as an “in situ vaccine” due to its stimulation of the adaptive immune system [35] and its induction effects in the setting of immune checkpoint inhibitors [36]. However, the conventional, daily fractionated RT required for tumor control may sterilize the immunostimulatory microenvironment or cause systemic lymphopenia and thus abrogate the vaccine effect. By delivering a higher dose per fraction, SBRT can extend the time between fractions without adverse tumor control and stimulate the immune system [37,38,39]; however, the optimal timing between fractions to preserve innate immune function remains under investigation. In a murine model, we discovered that improved tumor control was achieved with anti-PD-L1 therapy in conjunction with the prolonged spacing of radiation fractions of 10 days apart, as opposed to daily or at 4-day intervals, in both immunologically cold and hot tumors [40].

We hypothesize that this fractionation schema will work more synergistically with immunotherapy in the clinic, and we have launched several early-phase clinical trials exploring this new paradigm [40]. We have integrated adaptive radiation therapy into this concept, which is an emerging modality whereby radiation fields are adjusted in a real-time fashion to generate a dose distribution indexed to day-of anatomy, rather than anatomy at time of simulation, and we have termed this as “personalized, ultrahypofractionated stereotactic adaptive radiation therapy” (PULSAR). In this paradigm, a pulse of radiation (often ≥ 5 Gy) is delivered every 2–4 weeks for 2–5 pulses, dependent on normal tissue toxicity, timed within 48 h of checkpoint-inhibitor infusions. The targets are adapted at each pulse on CT- or MR-guided online adaptive platforms or in stereotactic radiosurgery.

PULSAR is thus able to deliver ablative doses in conjunction with checkpoint inhibitors, without a pause in systemic therapy, allowing for an immune response and adaptation of radiation fields based upon tumor response (Figure 2). Fields can be modified to account for response, or progression, of the tumor, and the dose can also be modified per pulse, dependent upon the clinical scenario. Furthermore, extended time between fractions allows for normal-tissue repair and thus decreased acute toxicity, likely improving quality of life, an endpoint explored in the aforementioned trials. This paradigm is enticing, particularly in radioresponsive tumors where a significant change in size may be expected after a few pulses of radiation. In the setting of treatment-naïve metastatic disease, the expeditious initiation of systemic therapy with PULSAR can provide local and systemic control concurrently and, possibly, improve the response to immunotherapy by enriching antigen expression and thus the activation of tumor-specific CD8+ T cells.

Specifically, a technique such as PULSAR may display efficacy in metastatic disease since the radiotherapy pulses release a variety of tumor-specific antigens. In turn, this stimulates adaptive immunity to react to and further recognize systemic disease, such as through production of high-affinity tumor antibodies or expansion of tumor-specific immune cell receptors [41]. To maximize the adaptive response, a potential strategy may be to target multiple metastatic lesions, as associated antigens are heterogenous and can therefore enhance immune responsivity and stimulate adaptive immune response and priming via different mechanisms.

PULSAR has also shown promise in stimulating robust immunological memory. When PULSAR-treated tumor-free mice were “re-challenged” with tumor cell injections, an immediate response was observed leading to rejection. Notably, this was not observed in mice that were treatment-naïve when re-implanted with the same tumor cells. This appears to be mediated by antigen-specific memory CD8+ T cells, of which a population may remain for several months following initial exposure and priming. While the mechanism of this boost in immunological memory remains unclear, it may be due to the presence of tissue-resident memory T cells or lymphocytic cells that do not recirculate. One study found a certain subset of T cells in irradiated lesions had increased motility and IFN-gamma levels, as well as transcriptional patterns similar to those in tissue-resident memory cells. It is likely that these are the cells that have survived irradiation, and indeed, they may act as a main stimulator of immune response thereafter. In PULSAR-based treatment, these cells may be even more abundant due to the pulsed radiotherapy approach that allows for less normal tissue toxicity and may avoid the pitfall of lymphopenia seen in the traditional fractionation schema discussed earlier [42].

Aside from animal models, the pulsed and extended-time radiotherapy approach has been validated by several studies. In a patient with metastatic renal cancer, hilar mass volume decreased from 68 cc to 3 cc following a PULSAR-based approach with 12 Gy pulses each spaced 1 month apart [43]. The effect of longer spacing between fractions has also been interrogated in other studies. The PATRIOT trial demonstrated that patients with intermediate-risk prostate cancer had improved acute bowel and urinary toxicity without compromising PSA control when irradiated once weekly compared to every other day [44]. This indicates that, in addition to the benefits of utilizing a personalized approach in adapting to treatment-related changes in each pulse by taking anatomy and tumor response into account between treatments, PULSAR-based therapy also possesses the unique advantage of allowing for more normal tissue recovery and thus avoiding the full extent of the adverse side effects observed with more typical treatments. A randomized study of SBRT for lung cancer found that patients irradiated twice weekly had a lower incidence of post-SBRT dyspnea compared to those treated daily, furthering adding credibility to the approach of avoiding radiation-related toxicity by expanding the intervals between treatments [45].

It should be noted that there is little concern that extended periods of time between fractions are detrimental due to tumor progression. SBRT treatments showed negligible changes in terms of tumor progression up to months after or between treatments, even at doses as low as 8 Gy [43]. Thus, with no discernable detriment to tumor control, and with an observed benefit in regard to adverse treatment-related effects, extending the time between pulses may be of further benefit. This approach may move the field towards better-tolerated treatments, with the ability to further personalize and adapt each treatment to patient anatomy and tumor change, and to optimize the immunostimulatory effects of highly ablative doses by retaining needed actors in the tumor–immune compartment.

## 6. Closing Remarks and Future Directions

When compared with conventionally fractionated radiation, the ablative potential of SBRT, along with fractionated regimens that involve few treatments spaced several days apart, may support an increase in tumor-antigen presentation, the enhancement of immunostimulatory components, and an improvement in tumor-microenvironment immune-cell infiltration. The approach of utilizing SBRT in temporally spaced treatments avoids the attenuation of critical, anti-tumor immune components that are typically observed with daily fractionated treatments utilizing conventional radiation schema. Utilizing highly ablative doses in this manner leads to immunoactivation and enhanced recognition, regardless of PD-1 and PD-L1 status. Several studies now support the importance of combining SBRT with immune checkpoint blockade and, more importantly, timing of optimal tumor–immune cell priming. Such studies indicate that ablative doses of radiation delivered prior to systemic immunotherapy may be the most promising path forward. Finally, the advent of PULSAR-based therapy, whereby ablative doses are delivered in pulses of treatment that are several weeks apart in some instances, combined with personalized and adaptive approaches to tumor changes across time, may be beneficial for several reasons. These benefits include the ability to optimize direct tumor control via the assessment of tumor size and location with each delivered pulse and the further potentiation of immune recognition through combination with concurrent immune checkpoint blockade. Further studies are needed to assess the effect of the length of time between pulses on tumor response in combination with immunotherapy and the effect of dosing and pulse sequencing on tumor microenvironmental changes and the recruitment of immunostimulatory components.

## Figures and Tables

**Figure 1 cancers-14-04530-f001:**
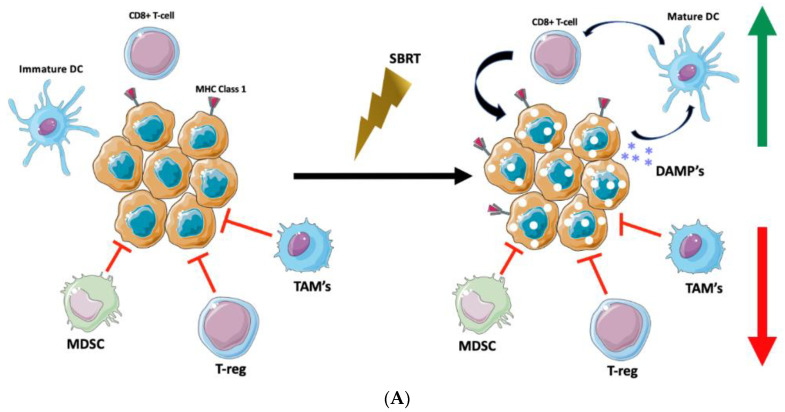
Targeting the Tumor Microenvironment with SBRT. (**A**) Stereotactic Body Radiotherapy offers delivery of highly potent ablative doses of radiation which stimulate expression of MHC class 1 molecules for enhanced tumor recognition, release of tumor damage-associated molecular patterns (DAMP’s) which may lead to maturation of dendritic cells, stimulation of cytotoxic T-cell activity via dendritic-cell antigen presentation, and enhanced anti-tumor immunomodulation. (**B**) Figure Legend.

**Figure 2 cancers-14-04530-f002:**
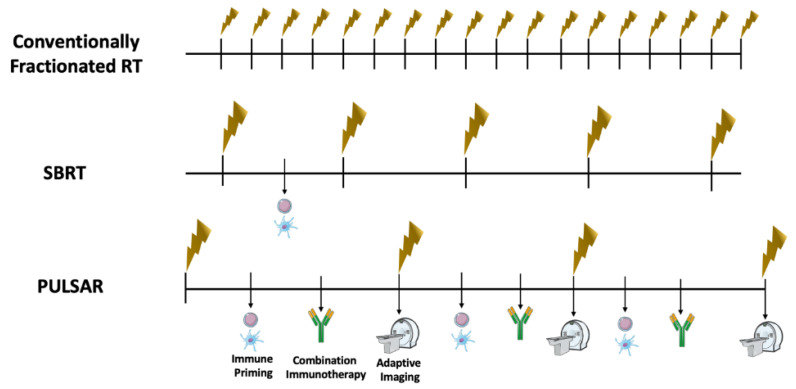
Potential Advantages of PULSAR Compared to Traditional Radiation Schema. Conventionally Fractionated radiation is characterized by daily treatments of non-ablative doses of radiation, that rarely lead to sufficient immune-stimulatory effects. Stereotactic Body Radiotherapy has the potential to offer ablative doses of radiation in 1 to 5 fractions, and may lead to sufficient immune priming to enhance anti-tumor immune response. However, with this approach there is no adaptation to changes in tumor anatomy between treatments, and considering a regular schedule of treatment that is typically every other day there is rarely combination with immunotherapy in between treatments. In personalized ultra-hypofractionated adaptive radiotherapy (PULSAR), pulses of ablative treatment can be delivered 2 weeks or even 1 month apart, with immune priming and immunotherapy administration regularly occurring between pulses to synergize for optimal anti-tumor effect. Furthermore, imaging prior to each pulse is acquired for personalized adaptation to tumor changes in anatomy over time.

## Data Availability

No data reported.

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
