# Peer review of "The Evolving Interplay of SBRT and the Immune System, along with Future Directions in the Field"

_cancers, 2022, doi:10.3390/cancers14184530_

Round 1
Reviewer 1 Report
This article is indicated by the Authors as a “review” but, not having the methodological principles of a review, it should be considered a "commentary article", as exactly listed by Editors.
As such it represents a focus on a series of related issues, which concern:
1) the immune-stimulatory effects of stereo body;
2) the timing of radiotherapy, in association with immune checkpoint blockade;
3) the use of ultrafractionated stereotactic radiotherapy combined with adaptive treatment and in combination with immune-checkpoint inhibitors.
The topics are discussed in separate sections, with references to preclinical and clinical experiences from the literature, to support the information provided.
The bibliography is however relevant and well inserted.
The clear, even if simple, iconography is accompanied by a descriptive caption, with concepts that could be part of the text.
The work could reach a more defined level of interest for the reader, with the following tips:
- not defining the work as a review.
- identifying a main topic among those treated.
- inserting keywords for a more focused search.
- creating a section, after the introduction, on evidences of immune-modulatory activity of radiotherapy, in relation to timing of radiation and its fractionation and in relation to combination with immune checkpoint inhibitors, separating these concepts from subsequent sections.
Therefore the article can be improved.
Author Response
Thank you very much for your review and kind words regarding our manuscript submission. We have made adjustments of the commentary based on your review of our submission in order to further enhance this work. In the abstract submission, we have defined the work as a commentary instead of a review for clarification to all readers. We have identified and emphasized in our introduction that the main topic of the manuscript will center of the immune-modulatory effect of SBRT, with the other relevant subsections discussing these areas within that context. We have taken your comments and that of reviewer 2 into account, in expanding the first section following the introduction to include further cited evidence of immune-modulatory activity of radiation at the cellular level to give additional context to the subsequent sections as they relate to timing and adaptive therapy. Finally, we are including keywords in our manuscript to allow for a more focused search by readers.
Reviewer 2 Report
It may help to redistribute section 2 into paragraphs within the remaining sections. Some of the cited data can be placed in the introduction without specific reference to fractionation, while the ones supporting various fractionation schedules can be placed into section 3 (which would be in regards both timing and fractionation).
There should be more development of the biological mechanisms of SBRT as an immune modulator in section 2, including and the intra and extracellular signaling involved (cytokines, immune populations apart from T cells - which are mentioned in the figure and in some of the texts but may be more comprehensively developed, as well as the role of intracellular organelles in the DAMPs (nuclear/mitochondrial dna damange, oxidative damage, CGAS/STING, various receptors, etc). Some of this is discussed in the first few sections of section 3 and can be moved into section 2. This would setup the biological rationale for the next section focusing on how timing and dose of SBRT may change the immune response.
Section 3 does well to include both theoretical and empirical evidence regarding timing and fraction size and but it is not clear from the evidence that there is some conflicting evidence regarding both timing and fraction size in the literature so additional data should be included to make this clear.
Section 4 motivates pulsar well but it is has a few redundancies (mentions the adaptive nature of pulsar in both paragraph 2 and 3). Additionally there are multiple other avenues of altered radiotherapy delivery that are being explored (namely FLASH, but also spatially fractionated therapy).
Author Response
Thank you for taking the time to review our manuscript, the provided suggestions on our invited commentary, and your kind words regarding our submission. We have taken these suggestions into account to include further biological mechanisms of SBRT as an immune-modulatory treatment modality in the section following our introduction, where this is discussed in detail, and have moved the discussion of some of these mechanisms from section 3 to section 2 as suggested. We have also worked to reduce the redundancy in descriptions of PULSAR in sections 2 and 3 prior to section 4. Finally, we have included further nuance regarding timing of radiation with immunotherapy in section 3, taking care to establish that it is still not clear what optimal timing of these approaches are and that debate still exists in the field that warrants further investigation. With our institutional experience utilizing PULSAR as well as production of prior publications on this topic, we have limited the scope of this invited commentary to PULSAR as the future direction discussed. With limited institutional experience in FLASH and other modalities, we have chosen not to provide expert commentary on these alternative avenues that may warrant separate manuscripts dedicated to these topics.
Reviewer 3 Report
The authors have done great job by summarizing the role of SBRT and immune system and how it can be novel treatment in targeting the tumors.
1. The content is good with significant presentation and scientific soundness.
2. Manuscript should be accepted after the final spell and grammar check.
Author Response
Thank you for your kind words regarding our manuscript, we look forward to the next review of our submission.
Round 2
